# Zero-shot Faithfulness Evaluation for Text Summarization with Foundation Language Model

**Qi Jia**     **Siyu Ren**
Shanghai Jiao Tong University, China
{Jia_qi, roy0702}@sjtu.edu.cn

**Yizhu Liu**
Meituan, China
liuyizhu@meituan.com

**Kenny Q. Zhu**[*]
University of Texas at Arlington, USA
kenny.zhu@uta.edu

## Abstract

Despite tremendous improvements in natural language generation, summarization models still suffer from the unfaithfulness issue. Previous work evaluates faithfulness either using models trained on the other tasks or in-domain synthetic data, or prompting a large model such as ChatGPT. This paper proposes to do zero-shot faithfulness evaluation simply with a moderately-sized foundation language model. We introduce a new metric FFLM, which is a combination of probability changes based on the intuition that prefixing a piece of text that is consistent with the output will increase the probability of predicting the output. Experiments show that FFLM performs competitively with or even outperforms ChatGPT on both inconsistency detection and faithfulness rating with 24x fewer parameters. FFLM also achieves improvements over other strong baselines.

## 1 Introduction

Faithfulness evaluation for text summarization aims at measuring if the information in a summary is fully covered by and consistent with the source document [1]. Although automatic text summarization has achieved remarkable improvements with pre-trained language models (Zhang et al., 2020; Lewis et al., 2020; Liu et al., 2021, 2022a; Zhang et al., 2023) in recent years, especially in the aspect of fluency and informativeness. However, these neural models tend to generate unfaithful summaries. An effective faithfulness evaluation metric not only helps for implementing summarization systems in real applications but also plays a key role in developing more faithful summarization models, such as by data filtering (Matsumaru et al.,

---

[*] The corresponding author.

[1]We use the words "faithfulness", "consistency" and "(without) hallucination" interchangeably. Extrinsic hallucinations that are correct to the world knowledge are regarded as unfaithfulness in this work.

2020) or doing post-hoc corrections (Chaudhury et al., 2022).

Most previous work for faithfulness evaluation either takes advantage of models trained on related tasks for zero-shot evaluation (Goodrich et al., 2019; Falke et al., 2019; Wang et al., 2020), or does weakly-supervised evaluation with synthetic in-domain data (Kryściński et al., 2020). The former requires transferring out-of-box models to the summarization domain (Mishra et al., 2021), which lacks guarantees on the models' performance and suffers from error propagation (Ji et al., 2023). The latter one shows poor generalization ability (Laban et al., 2022) as a result of the limited synthetic rules that couldn't cover various kinds of hallucinations. Recently, as ChatGPT (OpenAI, 2022) has shown amazing generation abilities on various tasks, researchers attempt to do human-like evaluation by designing prompts to query the model in the zero-shot manner (Luo et al., 2023). However, such strong language models are still sensitive to nuances, showing unstable performance with different wording of prompts (Gao et al., 2023; Chen et al., 2023).

Considering the above weaknesses, we think that an ideal faithfulness evaluation metric for summarization should be independent of other tasks and dataset-specific expertise, be able to generalize among different benchmarks and robust for the same document-summary pair. Zhou et al. (2023) concludes that instruction tuning is just to teach the model to produce high-quality output while almost all of the knowledge has been learned during pre-training for large language models. Based on their findings, we wonder: can we get rid of the popular prompting approaches and calculate the faithfulness score simply with a foundation language model, which meets the above expectations?

In this work, we propose a metric named FFLM for zero-shot faithfulness evaluation with a foundation language model. The intuition behind FFLM

is that the generation probability of a piece of text will increase when prefixing another piece of consistent text. Following this intuition, we classify different kinds of probability changes into changes with prior probability and changes with conditional probability. The former contains a comparison between the vanilla sequence-to-sequence probabilities of the summary given document and unconditional probabilities of the summary, and a similar comparison by changing the position of the document and the summary. The latter calculates the vanilla sequence-to-sequence probability with another conditional probability by adding a piece of prefix text. Similar intuition has been considered in previous works (She et al., 2023; Son et al., 2022). The major differences are that their metrics were carried out on models fine-tuned by summarization data and they only consider a single kind of probability changes. Our FFLM is based on the foundation language model, and we hypothesize that these different probability changes capture different hallucinations (see Sec. 4.4) which should be considered as a whole.

On top of these three components of probability changes, we introduce a feasible design of FFLM by re-weighting each token and each component to get the final faithfulness score. We did experiments in both the inconsistency detection setting and the faithfulness rating setting for summarization evaluation. The results show the favorable performance of our FFLM across different settings and datasets [2]. Our contributions are as follows:

- We propose to do zero-shot faithfulness evaluation based on a foundation language model(Sec. 4.6).

- We introduce a comprehensive evaluation metric FFLM by calculating the probability changes of the desired output in different ways(Sec. 2) and verify the rationality of our metric design(Sec.4.3).

- Experiments on different evaluation settings show that FFLM based on LLaMa with only 7 billion parameters can achieve competitive performances or even outperforms ChatGPT among different datasets(Sec 4.1 and 4.2).

[2]The code and dataset for this paper are available at https://github.com/JiaQiSJTU/FaithEval-FFLM.

## 2 Approach

Given a source document $X = \{x_1, ..., x_n\}$ and the corresponding summary $Y = \{y_1, ..., y_m\}$, the goal of this work is to design a metric FFLM measuring the faithfulness of $Y$ based on the foundation model $LM(\cdot)$. We adopt $LM(\cdot)$ under the teacher-forcing strategy, which can provide a sequence of generation probabilities $p$ of a given text with or without other conditional inputs. We first introduce three probability changes for faithfulness measurements and then propose a feasible design of our comprehensive metric FFLM. Scores proposed by She et al. (2023) and Son et al. (2022) are in Appendix A.

### 2.1 Faithfulness Measurements via Probability Changes

The intuition is that the generation probability of a piece of text will increase when providing more related and consistent information. On the contrary, the generation probability will drop when conditioned on inconsistent information. Accordingly, we considered three different probability changes in two categories as follows.

**Changes with Prior Probability:** The prior probability of $Y$ can be estimated by the foundation model $LM(\cdot)$:

$$p_Y^{lm} = LM(Y) = \{p_{y_i}^{lm}\}|_{i=1}^m \qquad (1)$$

and the sequence-to-sequence probability of $Y$ given $X$ is:

$$p_Y^{s2s} = LM(Y|X) = \{p_{y_i}^{s2s}\}|_{i=1}^m \qquad (2)$$

If $Y$ is a faithful summary, the sequence-to-sequence probability $p_Y^{s2s}$ should be larger than the prior probability $p_Y^{lm}$ as more information consistent to $Y$ is given by conditioning on $X$. Therefore, a faithfulness measurement can be defined as:

$$\Delta_{p_Y}^{prior} = \frac{1}{m} \sum_{i=1}^m p_{y_i}^{s2s} - p_{y_i}^{lm} \qquad (3)$$

From another point of view, we expect that the generation of $Y$ highly relies on $X$, instead of parametric knowledge stored in $LM$ which is a main resource of hallucinations (Ji et al., 2023).

Similarly, a faithful $Y$ can support the contents in $X$. Thus, the differences between the sequence-to-sequence probability of $X$ given $Y$ and the prior

probability of $X$ is another reasonable measurement:

$$p_X^{lm} = LM(X) = \{p_{x_i}^{lm}\}|_{i=1}^n$$
$$p_X^{s2s} = LM(X|Y) = \{p_{x_i}^{s2s}\}|_{i=1}^n \quad (4)$$
$$\Delta_{p_X}^{prior} = \frac{1}{n}\sum_{i=1}^n p_{x_i}^{s2s} - p_{x_i}^{lm}$$

**Changes with Conditional Probability:** Instead of comparing sequence-to-sequence generation probabilities with prior probabilities, another way is to add more information $P$ besides the input document $X$, leading to an influence on the generation probability of $Y$. Following She et al. (2023), we simply set $P = Y$. In this way, if $Y$ is inconsistent with $X$, prefixing $P$ will cause information contradictions in the input and decrease the probability of $Y$ compared to a consistent one. Mathematically, the third measurement is:

$$p_Y^{pref} = LM(Y|P,X) = \{p_{y_i}^{pref}\}|_{i=1}^m$$
$$\Delta_{p_Y}^{cond} = \frac{1}{m}\sum_{i=1}^m p_{y_i}^{s2s} - p_{y_i}^{pref} \quad (5)$$

We didn't consider $X$ and $Y$ reversely here. The main reason is that inputting the sequence $[P = X, Y, X]$ to $LM(\cdot)$ is much more costly and may exceed the max sequence length of most models since $X$ is much longer than $Y$, i.e., $n \gg m$.

## 2.2 A Feasible Design of FFLM

Goyal et al. (2022) found that high-loss tokens generally correspond to unfaithful contents during training a summarization model. Inspired by this finding and the success of the loss truncation training algorithms (Kang and Hashimoto, 2020), we think that more attention should be paid to such high-loss (or low-probability) tokens when calculating the faithfulness scores. So, instead of simply averaging the probability changes to get the final score for an (X, Y) pair, we adopt two operations. First, we take the logarithm of the probabilities before subtraction, which will magnify changes on the low-probability tokens. Second, we re-weight each token based on $p_Y^{s2s}$ and $p_X^{s2s}$ correspondingly. We get:

$$\Delta_Y^{prior} = \frac{1}{m}\sum_{i=1}^m e^{p_{y_i}^{s2s}}(\log p_{y_i}^{s2s} - \log p_{y_i}^{lm})$$
$$\Delta_X^{prior} = \frac{1}{n}\sum_{i=1}^n e^{p_{x_i}^{s2s}}(\log p_{x_i}^{s2s} - \log p_{x_i}^{lm})$$
$$\Delta_Y^{cond} = \frac{1}{m}\sum_{i=1}^m e^{p_{y_i}^{s2s}}(\log p_{y_i}^{s2s} - \log p_{y_i}^{pref})$$
$$(6)$$

Finally, FFLM is a combination of these metrics:

$$\text{FFLM} = \alpha\Delta_Y^{prior} + \beta\Delta_X^{prior} + \delta\Delta_Y^{cond} \quad (7)$$

where $\alpha$, $\beta$, and $\delta$ are weighting parameters in the range of 0 to 1 and $\alpha + \beta + \delta = 1$. These three weights can be tuned on a validation set, or set manually as hyper-parameters.

## 3 Experiment Setup

We present two evaluation settings considered by previous work for faithfulness evaluation first, with the implementation details of FFLM for them later.

### 3.1 Inconsistency Detection

Inconsistency detection regards the faithfulness evaluation as a binary classification problem. In other words, human annotators or automatic metrics only need to recognize whether the summary is faithful to the document or not.

***Datasets:*** The SUMMAC Benchmark (Laban et al., 2022) is a benchmark consisting of six summarization evaluation datasets, including Co-GenSumm Falke et al. (2019), SummEval (Fabbri et al., 2021), FRANK (Pagnoni et al., 2021), Polytope (Huang et al., 2020), FactCC (Kryściński et al., 2020) and XSumfaith (Maynez et al., 2020). It standardized these datasets by changing their original labels into a binary label and split each dataset into a validation set and a test set. Most of the original datasets are labeled by three or more annotators, except Polytope and FactCC.

***Evaluation Metric:*** Balanced accuracy (Brodersen et al., 2010) is adopted as the primary evaluation metric, which requires binary labels for computation. For approaches with continuous scores, a threshold can be selected via the validation set.

***Baselines:*** We borrowed the baselines from Laban et al. (2022)'s work, including linguistic feature-based metrics NER-Overlap (Laban et al., 2021) and DAE (Goyal and Durrett, 2020), NLI-based metric MNLI-doc (Kryściński et al., 2020) and SUMMAC$_{ZS}$ (Laban et al., 2022), QA-based metrics FEQA (Durmus et al., 2020) and QuestEval (Scialom et al., 2021), prompting with Chat-GPT (OpenAI, 2022) [3], and two weakly-supervised baselines FactCC-CLS (Kryściński et al., 2020)

---

[3]As Chen et al. (2023) shows that faithfulness evaluation is less reasoning-intensive and chain-of-though (Wei et al., 2023) prompting even hurts performances, we only compared with ChatGPT using a vanilla prompt.

| Setting | Dataset | Val | Test | Source |
|---------|---------|-----|------|--------|
| | CoGenSum | 1281 | 400 | C |
| Inconsistency | SummEval | 850 | 850 | C |
| Detection | FRANK | 671 | 1575 | C+X |
| (SUMMAC | Polytope | 634 | 634 | C |
| Benchmark) | FactCC | 931 | 503 | C |
| | XSumFaith | 1250 | 1250 | C |
| | FRANKCNN | - | 1250 | C |
| | QAGSCNN | - | 235 | C |
| Faithfulness | SummEval | - | 1600 | C |
| Rating | FRANKXSUM | - | 996 | X |
| | QAGSXSUM | - | 239 | X |

Table 1: Statistics of the datasets. "C" and "X" are short for CNN/DM (Nallapati et al., 2016) and XSum (Narayan et al., 2018) respectively.

and SUMMAC$_{CONV}$ (Laban et al., 2022). Besides, we implemented the language modeling-based metric BARTScore (Yuan et al., 2021) and metrics based on probability changes include CoP (She et al., 2023) and HaRiM (Son et al., 2022). These three metrics were suggested to use the CNN/DM (Nallapati et al., 2016) fine-tuned BART model [4] for calculation. We also improved the latter two metrics with our proposal by calculating with a foundation language model, LLaMa, for comparisons.

## 3.2 Faithfulness Rating

Faithfulness rating defines the evaluation as a Likert scale coring problem. Annotators or metrics score each summary according to its faithfulness. Generally, the higher, the more faithful.

***Datasets***: Following Son et al. (2022), we experimented on five different datasets: FRANKCNN and FRANKXSUM from Pagnoni et al. (2021), QAGSCNN and QAGSXSum from Wang et al. (2020), and SummEval (Fabbri et al., 2021). For the first four datasets, human judgments were originally done on the sentence level. The faithfulness rating of the whole summary is collected by doing majority voting on each summary sentence among annotators and averaging among sentences. SummEval contains human scores in the range of 1 to 5 in the aspect of consistency. More details are in Table 1.

***Evaluation Metrics***: Pearson($\gamma$), Spearman($\rho$), and Kendall($\tau$) correlation coefficients are used to measure the alignments between faithfulness ratings annotated by annotators and automatic metrics. The correlations are the higher the better. We consider the summary-level correlations for all datasets. Besides, system-level correlations are calculated

---
[4] https://huggingface.co/facebook/bart-large-cnn

on SummEval which contains annotations for 16 extractive or abstractive summarization models.

***Baselines***: Rouge-2 F1 (Lin, 2004), Meteor (Banerjee and Lavie, 2005), BLEU (Papineni et al., 2002) and BERTScore F1 (Zhang et al., 2019a) are widely-accepted summarization evaluation metrics. We report their best results in Son et al. (2022) by calculating between the summary and the source document. QAGS (Wang et al., 2020) is another QA-based metric. Others are the same as the ones for inconsistency detection.

## 3.3 Implementation Details

We implemented FFLM with the foundation language model LLaMa (Touvron et al., 2023). It contains models with different sizes, where LLaMa7b is selected for our main experiments. We add "TL;DR" between the conditional sequence and the target sequence. The weights in Eq. 7 are determined in $\{0.0, 0.1, ..., 1.0\}$ according to the performance on the corresponding validation set for inconsistency detection. For faithfulness rating, we set $\alpha, \beta, \delta$ as $0.25, 0.25, 0.5$ respectively, with the intuition that the former two are from the same category as introduced in Sec. 2.1. Our experiments are done on a single RTX 3090.

# 4 Results and Analysis

This section includes the main results for inconsistency detection and faithfulness rating, together with an ablation study, an analysis of error types, and comparisons of different model sizes of our FFLM. We also discussed our metric and the prompting approach with or without instruction tuning under the same model size.

## 4.1 Performance on Inconsistency Detection

The results on inconsistency detection are in Table 2. Our proposed metric FFLM achieves state-of-the-art performance on 3 datasets including Co-GenSum, SummEval, and FRANK, and outperforms ChatGPT on 5 out of 6 datasets from the SUMMAC benchmark except XSumFaith.

Both Polytope and FactCC are only labeled by a single annotator. As a result, their labels may not be as convincing as the other datasets. Although QuestEval, the best QA-based metric, achieves the top-1 accuracy on Polytope, it performs mediocrely on the rest. The weakly-supervised baselines FactCC$_{CLS}$ and SummaC$_{Conv}$ are trained with synthetic data constructed with human expertise

| Metric | CoGenSum | SummEval | FRANK | Polytope | FactCC | XSumFaith |
|---|---|---|---|---|---|---|
| NER Overlap | 53.0 | 56.8 | 60.9 | 52.0 | 55.0 | 63.3 |
| MNLI-doc | 57.6 | 66.6 | 63.6 | 61.0 | 61.3 | 57.5 |
| $FactCC_{CLS}$ | 63.1 | 60.1 | 59.4 | 61.0 | 75.9 | 57.6 |
| DAE | 63.4 | 70.3 | 61.7 | 62.8 | 75.9 | 50.8 |
| FEQA | 61.0 | 53.8 | 69.9 | 57.8 | 53.6 | 56.0 |
| QuestEval | 62.6 | 72.5 | 82.1 | **70.3** | 66.6 | 62.1 |
| $SummaC_{ZS}$ | 70.4 | 78.7 | 79.0 | 62.0 | 83.8 | 58.4 |
| $SummaC_{Conv}$ | 64.7 | 81.7 | 81.6 | 62.7 | **89.5** | **66.4** |
| BARTScore | 62.5 | 66.7 | 80.2 | 57.3 | 68.4 | 56.9 |
| $CoP_{BART}$ | 65.3 | 63.9 | 77.7 | 60.0 | 69.0 | 61.5 |
| $HaRiM_{BART}$ | 58.9 | 76.6 | 81.8 | 55.8 | 67.3 | 56.2 |
| ChatGPT | 63.3 | 76.5 | 80.9 | 56.9 | 74.7 | 64.7 |
| $CoP_{LLaMa}$ | 65.4 | 83.6 | 83.1 | 55.4 | 78.6 | 54.1 |
| $HaRiM_{LLaMa}$ | 57.1 | 80.0 | 83.4 | 58.8 | 69.8 | 53.4 |
| FFLM | **71.8** | **84.4** | **83.9** | 61.5 | 77.3 | 58.9 |

Table 2: Balanced accuracy(%) on the SUMMAC benchmark. The best result for each dataset is in bold. Scores of FFLM better than other metrics based on the foundation model are underlined.

| Metric | FRANKCNN | | | QAGSCNN | | | SummEval | | | FRANKXSUM | | | QAGSXSUM | | |
|---|---|---|---|---|---|---|---|---|---|---|---|---|---|---|---|
| | $\gamma$ | $\rho$ | $\tau$ | $\gamma$ | $\rho$ | $\tau$ | $\gamma$ | $\rho$ | $\tau$ | $\gamma$ | $\rho$ | $\tau$ | $\gamma$ | $\rho$ | $\tau$ |
| Rouge-2 | 33.1 | 32.7 | 24.9 | 47.5 | 42.7 | 31.5 | 24.7 | 25.2 | 19.5 | 1.2 | 3.3 | 2.7 | 10.7 | 9.1 | 6.9 |
| Meteor | 23.0 | 22.9 | 17.4 | 27.7 | 32.4 | 23.4 | 14.3 | 12.2 | 11.2 | -0.5 | 0.5 | 0.4 | -1.5 | -7.1 | -5.2 |
| BLEU | 9.3 | 20.2 | 15.3 | 18.0 | 33.7 | 24.5 | 11.7 | 7.3 | 9.1 | -4.2 | -4.6 | -3.8 | 4.7 | -18.6 | -13.9 |
| BERTScore | 51.4 | 46.4 | 35.8 | 55.6 | 49.3 | 36.5 | 29.2 | 29.5 | 23.0 | 15.7 | 13.7 | 11.1 | -4.8 | -5.4 | -4.0 |
| $FactCC_{CLS}$ | 49.2 | 43.8 | 37.6 | - | - | - | 32.0 | 34.0 | - | 7.2 | 7.2 | 7.1 | - | - | - |
| FEQA | -1.8 | -1.0 | -0.8 | - | - | - | - | - | - | 2.6 | 0.8 | 0.6 | - | - | - |
| QAGS | 31.4 | 26.7 | 20.6 | 46.6 | 38.2 | 27.4 | 17.7 | 12.7 | - | -2.2 | -0.7 | -0.6 | 21.7 | 20.3 | 15.3 |
| QuestEval | - | - | - | 49.2 | 44.5 | - | 37.0 | 33.9 | - | - | - | - | 7.0 | 9.6 | - |
| DAE | 44.0 | 44.7 | 34.2 | - | - | - | 20.0 | 27.0 | - | 5.8 | 11.3 | 9.2 | - | - | - |
| BARTScore | 56.1 | 53.0 | 41.3 | 67.3 | 61.3 | 47.0 | 24.9 | 26.2 | 19.7 | 17.4 | 16.8 | 13.7 | 8.0 | 9.7 | 7.2 |
| $CoP_{BART}$ | 56.1 | 51.0 | 39.4 | 73.0 | 65.3 | 53.2 | 23.6 | 22.6 | 18.0 | 22.8 | 20.8 | 17.0 | 26.6 | 25.3 | 20.7 |
| $HaRiM_{BART}$ | 61.0 | 53.9 | 42.1 | 67.4 | 58.2 | 47.1 | 42.7 | 37.6 | 29.8 | 14.8 | 13.9 | 11.4 | 15.8 | 16.0 | 13.1 |
| ChatGPT | 50.0 | 46.0 | - | - | - | - | 49.0 | 35.0 | - | **34.0** | **27.0** | - | - | - | - |
| $CoP_{LLaMa}$ | 59.7 | 54.5 | 42.6 | **74.3** | **67.6** | **54.8** | 55.1 | 46.4 | 37.0 | 24.6 | 23.1 | 18.8 | 19.0 | 18.1 | 14.7 |
| $HaRiM_{LLaMa}$ | 56.9 | 51.9 | 40.3 | 68.6 | 60.0 | 48.2 | 56.1 | 45.5 | 36.4 | 18.6 | 16.7 | 13.6 | 9.1 | 10.0 | 8.2 |
| FFLM | **62.2** | **56.0** | **43.7** | 72.3 | 65.3 | 53.0 | **56.3** | **46.9** | **37.4** | 27.0 | 25.3 | 20.6 | **28.3** | **27.1** | **22.2** |

Table 3: Summary-level correlations(%) on the faithfulness rating datasets.

that may have certain similarities with the FactCC dataset. Therefore, $FactCC_{CLS}$ shows strong performance on the FactCC dataset while relatively weak on the others including datasets in Table 3, the same as the findings in Laban et al. (2022). Also, that's why $SummaC_{Conv}$ shows around 12% significant improvements over our FFLM.

Concentrating on the metrics based on probability changes, zero-shot metrics $CoP_{BART}$ and $HaRiM_{BART}$ perform not badly compared with previous SOTA $SummaC_{ZS}$, showing the potential of using probability changes for faithfulness evaluation. After introducing the foundation language model, their performances don't drop in most cases, indicating that fine-tuning with in-domain data is not necessary. However, the leading performance between these two metrics is unstable among datasets. $HaRiM_{LLaMa}$ outperforms $CoP_{LLaMa}$ on FRANK and Polytope, while on the rest datasets, the opposite is true. FFLM, as a comprehensive metric, successfully achieves improvements over both of them on 5 out of 6 datasets.

## 4.2 Performance on Faithfulness Rating

Summary-level results are in Table 3. The results of ChatGPT borrowed from Luo et al. (2023) show its inconsistency improvements among datasets: It doesn't exceed previous baselines on FRANKCNN, performs similarly on SummEval, and achieves conspicuous gains on FRANKXSUM. Besides, similar to the above analysis for comparisons among probability change-based metrics, our FFLM induces performance gains on 4 out of 5 datasets over $CoP_{LLaMa}$ and $HaRiM_{LLaMa}$, especially on datasets sourced from XSum. Unfortunately, FFLM still lags behind ChatGPT with 175 billion parameters on FRANKXSUM, showing ChatGPT's strong ability on dealing with highly abstractive summaries. This is also in line with ChatGPT's favorable performance on XSumFaith in Table 2. After all, FFLM achieves the best scores on FRANKCNN, SummEval, and QAGSXSUM, and performs competitively on the other datasets.

We also report the system-level results on SummEval in Table 5. FFLM performs similarly to ChatGPT according to the Spearman correlation.

| Metric | FRANKCNN | | | QAGSCNN | | | SummEval | | | FRANKXSUM | | | QAGSXSUM | | |
|---|---|---|---|---|---|---|---|---|---|---|---|---|---|---|---|
| | $\gamma$ | $\rho$ | $\tau$ | $\gamma$ | $\rho$ | $\tau$ | $\gamma$ | $\rho$ | $\tau$ | $\gamma$ | $\rho$ | $\tau$ | $\gamma$ | $\rho$ | $\tau$ |
| FFLM | **62.2** | **56.0** | **43.7** | 72.3 | 65.3 | 53.0 | **56.3** | **46.9** | **37.4** | 27.0 | 25.3 | 20.6 | 28.3 | 27.1 | 22.2 |
| *Ablations on the metric components* | | | | | | | | | | | | | | | |
| $\Delta_Y^{prior}$ | 32.0 | 26.4 | 20.1 | 11.9 | 7.5 | 5.7 | 24.9 | 20.5 | 16.1 | 20.0 | 19.2 | 15.6 | 20.9 | 21.3 | 17.4 |
| $\Delta_X^{prior}$ | 34.4 | 36.0 | 27.4 | 27.7 | 28.1 | 21.8 | 23.7 | 26.3 | 20.7 | 10.2 | 11.2 | 9.2 | 4.8 | 3.2 | 2.6 |
| $\Delta_Y^{cond}$ | 59.9 | 54.6 | 42.7 | 74.3 | 67.9 | 55.2 | 54.8 | 46.3 | 36.9 | 24.7 | 23.3 | 19.0 | 19.4 | 18.0 | 14.7 |
| $\Delta_Y^{prior}, \Delta_X^{prior}$ | 34.7 | 29.2 | 22.3 | 17.7 | 13.0 | 9.9 | 28.3 | 23.6 | 18.6 | 20.4 | 19.7 | 16.0 | 20.7 | 21.3 | 17.4 |
| $\Delta_Y^{prior}, \Delta_Y^{cond}$ | 61.0 | 54.2 | 42.3 | 68.1 | 60.2 | 48.4 | 54.4 | 45.1 | 36.0 | **28.3** | **26.5** | **21.6** | **29.4** | **28.6** | **23.4** |
| $\Delta_X^{prior}, \Delta_Y^{cond}$ | 60.3 | 54.7 | 42.6 | 73.9 | 66.5 | 54.0 | 54.7 | 46.6 | 37.1 | 24.7 | 23.2 | 18.9 | 19.8 | 18.3 | 14.9 |
| *Ablations on the metric designs* | | | | | | | | | | | | | | | |
| - w/o $w$ | 61.2 | 54.8 | 42.7 | 68.4 | 60.1 | 48.6 | 54.3 | 45.4 | 36.3 | 26.9 | 25.0 | 20.4 | 26.4 | 26.00 | 21.3 |
| - w/o log | 57.5 | 52.3 | 40.5 | 69.2 | 60.3 | 48.5 | 56.9 | 45.9 | 36.7 | 19.7 | 17.5 | 14.3 | 11.5 | 12.2 | 10.0 |
| - w/o $w$ and log | 56.3 | 51.3 | 39.6 | 66.5 | 57.6 | 46.4 | 54.4 | 45.0 | 36.0 | 18.8 | 16.6 | 13.5 | 11.0 | 12.4 | 10.2 |
| *Ablations on the combination weights ($\alpha, \beta, \delta$)* | | | | | | | | | | | | | | | |
| same | 60.9 | 54.0 | 42.1 | 67.5 | 58.6 | 47.1 | 54.1 | 44.9 | 35.8 | 28.3 | 26.4 | 21.5 | 29.2 | 28.7 | 23.5 |

Table 4: Ablations of FFLM on faithfulness rating. The highest scores are in bold.

| Metric | $\gamma$ | $\rho$ | $\tau$ |
|---|---|---|---|
| Rouge-2 | 50.0 | 59.9 | 68.8 |
| Meteor | 46.7 | 51.3 | 62.1 |
| BLEU | 45.0 | 28.7 | 62.1 |
| BERTScore | 68.3 | 68.0 | 86.8 |
| BARTScore | 30.1 | 25.9 | 18.3 |
| CoP$_{BART}$ | 19.9 | 35.9 | 25.0 |
| HaRiM$_{BART}$ | 75.9 | 61.2 | 45.0 |
| ChatGPT | - | 83.3 | - |
| CoP$_{LLaMa}$ | 88.3 | 81.8 | 63.3 |
| HaRiM$_{LLaMa}$ | 89.9 | **84.4** | **66.7** |
| FFLM | **90.4** | 83.2 | 65.0 |

Table 5: System-level correlations between metrics and human ratings on the SummEval dataset.

HaRiM$_{LLaMa}$ achieves a bit higher Spearman and Kendall correlation than FFLM, while FFLM performs better on Pearson correlation showing better linear correlation with human scores. Moreover, FFLM is more robust than HaRiM$_{LLaMa}$ on different evaluation settings considering the poor summary-level performance of HaRiM$_{LLaMa}$ especially for FRANKXSUM and QAGSXSUM in Table 3. Another observation is that although CoP and HaRiM backed on BART perform closely with them backed on LLaMa on the summary-level evaluation, they perform poorly on the system-level evaluation. This can be attributed to the fact that metrics based on CNN/DM fine-tuned BART have inductive bias (Son et al., 2022). They tend to prefer summaries generated by abstractive models, while extractive models are generally more faithful. Meanwhile, metrics based on the foundation language model don't show this bias, leading to the best results for ranking summarization systems.

To recap, our FFLM generalizes well among different task settings and different datasets, showing favorable performance over the baselines. It is backed on LLaMa with only 7 billion parameters and performs competitively with or even outperforms ChatGPT with 175 billion parameters, which

is much more efficient for faithfulness evaluation.

## 4.3 Ablation Study on Metric Designs

We carried out ablation studies of FFLM on faithfulness rating in Table 4. The ablation results on inconsistency detection are in Appendix B.

**Ablations on the metric components:** We test different combinations of the three probability changes. The results show that $\Delta_Y^{cond}$ is the most powerful component of FFLM. Its combination with $\Delta_Y^{prior}$ ranks first among ablations on both FRANKXSUM and QAGSXSUM. Together with $\Delta_X^{prior}$, our metric FFLM shows over 5% increases in Spearman correlation on QAGSCNN, 1.8% on FRANKCNN and SummEval, without much loss on the other two datasets, records more robust results. Moreover, combining different probability changes induces performance gains in most cases, reflecting the necessity of designing a comprehensive metric(More in Sec 4.4).

**Ablations on the metric designs:** We use $w$ and log to annotate the token-level weights and the logarithm operation introduced in Sec 2.2. Both operations contribute to the final FFLM, where log is more effective for datasets sourced from XSum and $w$ for the others.

**Ablations on the combination weights:** For the faithfulness rating task where we empirically set the weights $\alpha$, $\beta$ and $\delta$ as 0.25, 0.25 and 0.5, we compared it with the equaling weights, i.e., $\alpha = \beta = \delta = \frac{1}{3}$. FFLM performs relatively better.

## 4.4 Analysis on Error Types

By taking a look at the correlations between pairs of the metric components in Figure 6, we can see that the correlations vary among different datasets. None of the pairs show a high degree of correlation, indicating that these components may capture

| Metric | FRANKCNN | | | QAGSCNN | | | SummEval | | | FRANKXSUM | | | QAGSXSUM | | |
|---|---|---|---|---|---|---|---|---|---|---|---|---|---|---|---|
| | $\gamma$ | $\rho$ | $\tau$ | $\gamma$ | $\rho$ | $\tau$ | $\gamma$ | $\rho$ | $\tau$ | $\gamma$ | $\rho$ | $\tau$ | $\gamma$ | $\rho$ | $\tau$ |
| $\Delta_Y^{prior}, \Delta_X^{prior}$ | 44.7 | 46.2 | 32.0 | 32.5 | 35.8 | 23.8 | 29.5 | 34.4 | 23.7 | 28.3 | 31.1 | 20.9 | 28.0 | 24.9 | 17.0 |
| $\Delta_Y^{prior}, \Delta_Y^{cond}$ | 26.5 | 19.9 | 12.9 | 14.7 | 2.3 | 1.2 | 20.1 | 15.6 | 10.1 | 23.4 | 20.1 | 13.4 | -6.2 | -2.4 | -1.5 |
| $\Delta_X^{prior}, \Delta_Y^{cond}$ | 47.2 | 49.9 | 34.6 | 35.2 | 41.2 | 28.0 | 36.5 | 40.6 | 27.9 | 38.7 | 38.3 | 25.9 | -0.4 | 3.8 | 2.5 |

Table 6: Correlations between pairs of the metric components on faithfulness rating.

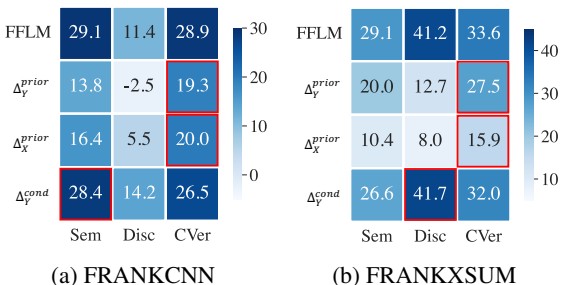

Figure 1: Spearman correlation(%) of different error types on FRANKCNN and FRANKXSUM. Highest correlations for each $\Delta$ is highlighted with red boxes.

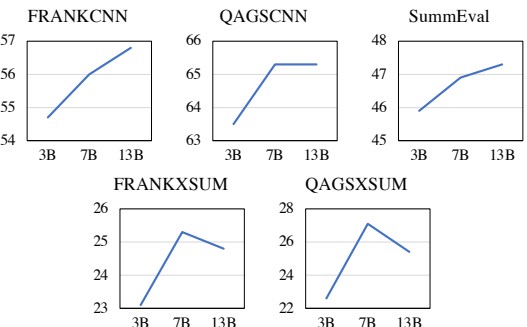

Figure 2: Spearman correlation(%) of FFLM with different model sizes on faithfulness rating datasets.

unfaithfulness from different aspects.

To figure out if different probability changes correlate well with different error types in the generated summaries, we take advantage of labels in the FRANKCNN and FRANKXSUM datasets. Pagnoni et al. (2021) divides the factual errors in the generated summaries into three groups. Semantic frame errors(**Sem**) include errors on the predicate, entities, and additional information about the circumstance. Discourse errors(**Disc**) consist of coreference errors and discourse link errors. Content verifiability errors(**CVer**) are closely related to extrinsic hallucinations (Ji et al., 2023), containing the out-of-article error and grammatical error. We randomly picked 50 error cases and 10 error cases for each error type from FRANKCNN and FRANKXSUM respectively, and mixed them with the rest faithful summaries. Spearman correlations averaged over 10 times are in Fig. 1.

We observed that $\Delta_Y^{cond}$ captures different errors best, which is accord with the ablation results in Table 4. Comparing among the scores for each $\Delta$ horizontally, we can see that the probability changes with prior probability is good at CVer errors on both datasets, and $\Delta_Y^{cond}$ at Sem errors or Disc errors. The differences among datasets reflect their different characteristics (Pagnoni et al., 2021). Summaries in FRANKCNN are made up of multiple sentences, resulting in more diverse and challenging situations for Disc errors than FRANKXSUM with single-sentence summaries. Thus, $\Delta_Y^{cond}$ increases dramatically from 14.2% on FRANKCNN

to 41.7% on FRANKXSUM for Disc.

FFLM made further improvements over $\Delta_Y^{cond}$ on both Sem and CVer, showing that combining different probability changes is reasonable and effective in most cases except Discs.

### 4.5 Performance on Different Model Sizes

To test FFLM's performance on different models sizes, we select LLaMa with 3 billion(3B), 7 billion(7B) and 13 billion(13B) parameters [5] that are trained on the same data volume with 1 trillion tokens and draw the diagram in Fig. 2 for faithfulness rating datasets. The scores consistently increase from LLaMa-3B to LLaMa-7B across the five datasets, while the improvements are not consistent for LLaMa-13B. Given a certain amount of data, increasing the number of parameters can enhance the model's language modeling ability and be helpful to faithfulness evaluation. On the other hand, when the model size keeps scaling up, more unexpected biases in the pre-training corpus may be memorized and will hurt the performance. This has also been pointed out by Ranaldi et al. (2023) and Nadeem et al. (2021).

In this way, we think that using larger foundation models may not be the best choice for faithfulness evaluation on summarization, which is also closely related to the research on investigating the optimal

---

[5]The corresponding checkpoints from hugging face are openlm-research/open_llama_3b, decapoda-research/llama-7b-hf, and decapoda-research/llama-13b-hf.

| Metric | FRANKCNN | | | QAGSCNN | | | SummEval | | | FRANKXSUM | | | QAGSXSUM | | |
|---|---|---|---|---|---|---|---|---|---|---|---|---|---|---|---|
| | $\gamma$ | $\rho$ | $\tau$ | $\gamma$ | $\rho$ | $\tau$ | $\gamma$ | $\rho$ | $\tau$ | $\gamma$ | $\rho$ | $\tau$ | $\gamma$ | $\rho$ | $\tau$ |
| *Prompting Approach* | | | | | | | | | | | | | | | |
| LLaMa-7B | -1.3 | -0.2 | -0.2 | 2.4 | 2.0 | 1.9 | 9.5 | 11.5 | 10.9 | 3.0 | 3.0 | 3.0 | -6.9 | -7.0 | -6.9 |
| Vicuna-7B | _17.6_ | _17.4_ | _15.5_ | _18.9_ | _19.8_ | _17.7_ | _13.1_ | _11.9_ | _11.1_ | _7.0_ | _5.5_ | _5.1_ | _10.2_ | _8.4_ | _8.0_ |
| Alpaca-7B | 5.4 | 6.8 | 6.2 | 10.2 | 9.3 | 8.7 | 3.0 | 6.5 | 5.6 | 3.8 | 3.4 | 3.4 | 2.3 | 1.4 | 1.4 |
| *Our Approach* | | | | | | | | | | | | | | | |
| LLaMa-7B | 62.2 | 56.0 | 43.7 | 72.3 | 65.3 | 53.0 | **56.3** | 46.9 | 37.4 | **_27.0_** | **_25.3_** | **_20.6_** | **28.3** | **_27.1_** | **_22.2_** |
| Vicuna-7B | **_62.7_** | **_56.7_** | **_44.3_** | **_73.1_** | **_67.2_** | **_54.6_** | 55.3 | **_47.2_** | **37.7** | 25.8 | 23.9 | 19.4 | 23.5 | 22.5 | 12.8 |
| Alpaca-7B | 61.4 | 55.3 | 43.2 | 71.4 | 66.0 | 52.6 | 55.8 | 47.1 | 37.6 | 26.2 | 24.6 | 20.0 | 24.2 | 25.4 | 20.7 |

Table 7: Comparisons with prompting and instruction-tuning techniques under the same model size. The highest correlations are in bold in each column and are underlined among each kind of approach.

model size and dataset size for training foundation language models (Hoffmann et al., 2022).

## 4.6 Comparisons with Prompting and Instruction-tuning

We compare our metric with prompting and instruction-tuning techniques under the same model size in Table 7 for faithfulness rating. Here, LLaMa-7B is the vanilla foundation language model. Vicuna-7B (Chiang et al., 2023) and Alpaca-7B (Taori et al., 2023) are initialized from LLaMa-7B and instruction-tuned with data collected in different ways. We present the maximum scores for each dataset among different prompts designed by previous works (Chen et al., 2023; Gao et al., 2023; Luo et al., 2023). The detailed prompts for each evaluation task are listed in Appendix C.

First, we observe that using models containing 7 billion parameters, FFLM outperforms the prompting approach across different models and datasets. The prompting results here lag behind the performance of ChatGPT dramatically. This leads to the conclusion that the effectiveness of prompting approaches relies highly on much larger models, while our metric FFLM can be a cheaper alternative with smaller models. Second, instruction tuning is important for improving the prompting approach, while is not necessary for our FFLM. It enhances the models' understanding ability on instruction templates in the prompts by further tuning with relatively small datasets. However, such manually collected datasets may contain unconscious bias and hurt FFLM's performance.

## 5 Related Work

### 5.1 Faithfulness Evaluation for Summarization

Faithfulness evaluation metrics can be classified into zero-shot ones and weakly-supervised ones.

Zero-shot evaluation metrics mainly take advantage of the models trained with related natural language tasks. Goodrich et al. (2019) adopted information extraction tools to extract the fact tuples from both the source document and the summary. Tuple mismatches reflect the hallucinations. The intuition behind question-answering-based metrics (Wang et al., 2020; Durmus et al., 2020; Scialom et al., 2021) is that identical answers should be generated when asking the same question to a summary and the corresponding document respectively. Natural language inference also shares commonalities with faithfulness evaluation in the way that information in a consistent summary should be entirely entailed by the source document (Falke et al., 2019; Mishra et al., 2021; Laban et al., 2022). However, all of these metrics highly rely on the domain-transfer ability of out-of-box models and suffer from error propagation.

Instead, weakly-supervised approaches choose to train classifiers by constructing synthetic in-domain data with heuristics by experts. Different kinds of inconsistency errors are simulated by perturbing the reference document-summary pairs (Kryściński et al., 2020; Utama et al., 2022; Yin et al., 2021). The limited heuristic makes it hard to cover all kinds of errors and shows poor generalization ability among datasets (Laban et al., 2022).

As language modeling-based metrics (Egan et al., 2022; Liu et al., 2022b) receive more attention, another small group of work for faithfulness evaluation computes probability changes with models fine-tuned on summarization datasets (She et al., 2023; Son et al., 2022; Xie et al., 2021), showing a biased preference for abstractive summaries. Based on this line of work, we propose FFLM based on the foundation language model. Our zero-shot metric doesn't require further training with in-domain or synthetic data and shows a strong generalization ability.

## 5.2 Evaluation with Large Language Models

With orders of magnitude more parameters and extensive training on large-scale data, large language models (LLMs) (Brown et al., 2020; Touvron et al., 2023) have exhibited surprising abilities that may not be observed in previous small language models. The strong capability in language comprehension naturally spurs research in exploring LLMs as better automatic evaluators for various text generation systems (Wang et al., 2023).

There are also some attempts of faithfulness evaluation by prompting large models (Luo et al., 2023) with different templates and strategies, such as adding detailed definitions (Gao et al., 2023) and chain-of-thought (Chen et al., 2023). None of these strategies achieve consistent improvements over the original prompt. Besides, neural models are sensitive to the choices of words (Chen et al., 2023), resulting in unstable performances(See Appendix D).

Our FFLM takes advantage of the strong capability of LLMs for faithfulness evaluation in a different way and shows competitive performance requiring a much smaller number of parameters than the well-known ChatGPT (OpenAI, 2022).

## 6 Conclusion

This paper focuses on zero-shot faithfulness evaluation for summarization and introduces a novel evaluation metric FFLM which is simply based on the foundation language model. Experiments on both the inconsistency detection benchmark and faithfulness rating datasets show the strong generalization ability of FFLM across various task settings and different datasets. It also shows favorable performance over strong baselines including ChatGPT. Using our proposed metric for more fine-grained consistency detection and designing more faithful summarization systems are future directions.

## Limitations

The main idea of this work is to do faithfulness evaluation based on a foundation language model by a combination of different probability changes. FFLM is just a feasible but not perfect metric design. Although it makes improvements over each $\Delta$ on almost all of the datasets in Table 4, it failed on the errors related to discourse errors on the FRANKCNN and FRANKXSUM dataset according to Fig. 1. Designing better aggregation metrics

based on specific analysis of different error types will be considered in the future.

Besides, in this work, our FFLM only calculates a single score for the whole summary without pinpointing the exact erroneous words or the specific error type. Considering the success of CoP (She et al., 2023) on token-level inconsistency detection and detailed inconsistency category evaluation, we hypothesize that our metric FFLM can be also used for these evaluation scenarios by adjusting the aggregation weights or combining it with the prompting approach.

Moreover, we limit our scope to faithfulness evaluation for text summarization in this paper because the definition of faithfulness evaluation for other generation tasks has some non-trivial differences. For example, the chit-chat utterances in dialogue generation (Dziri et al., 2022) are supposed to be acceptable under the evaluation for faithfulness, instead of being regarded as extrinsic hallucinations. The evaluation for sentence paraphrasing (Zhang et al., 2019b) should be bi-directional, i.e., the first sentence has to be consistent with the second one, and vice versa. We consider transferring FFLM with adjustment on the other tasks as future work.

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

## A  Preliminaries

CoP proposed by (She et al., 2023) is calculated based on $\Delta_{p_Y}^{cond}$. Although they only showed $\Delta_{p_Y}^{cond}$ in their paper, they took the logarithm of probabilities during implementation without analysis and explanations. Overall, CoP is:

$$\text{CoP} = \frac{1}{m}\sum\nolimits_{i=1}^{m}\log p_{y_i}^{s2s} - \log p_{y_i}^{pref} \quad (8)$$

HaRiM proposed by (Son et al., 2022) is derived from $\Delta_{p_Y}^{prior}$:

$$\text{HaRiM} = \frac{1}{m}\sum\nolimits_{i=1}^{m}(1-p_{y_i}^{s2s})[1-(p_{y_i}^{s2s}-p_{y_i}^{prior})] \quad (9)$$

Our FFLM is different from HaRiM in three ways: First, we propose to do faithfulness evaluation with foundation language models while they suggested to evaluate with models fine-tuned on summarization data. Second, HaRiM only considers one of the changes with prior probability, i.e., $\Delta_{p_Y}^{prior}$. Our metric not only adds a back constraint of $P(X|Y)$ additionally to HaRiM, but also considers the probability changes with conditional probability. Third, Specifically for the formula, the intuition of adding weights behind HaRiM and our FFLM are different regardless of the function-form variations. HaRiM's weight $(1-p_{y_i}^{s2s})$ was originally introduced to adjust the loss scale for training better neural machine translation models by Miao et al. (2021). FFLM adopted the weight $e^{p_{y_i}^{s2s}}$ and the logarithm to pay more attention to low-probability tokens which generally correspond to unfaithful contents according to (Goyal et al., 2022).

Xie et al. (2021) introduced another similar metric CoCo which is also based on probability changes. Instead of using the prior probability of $Y$ in $\Delta_{p_Y}^{prior}$, they condition $Y$ on the masked document $X'$ by removing $Y$-related information. The masking strategy is hard to design since locating the $Y$-related information is uneasy. Their complicated operation also largely slows down the inference speed.

We compared these three metrics backed on BART for faithfulness ranting in Table 8. Since CoCo's performances vary a lot with different masking strategies and none of them completely outperform the other two metrics on a single dataset, we didn't compare with it thoroughly in our work.

## B  Analysis on Inconsistency Detection

Ablation studies of FFLM for inconsistency detection are in Table 9. Fig. 3 illustrates the performances of FFLM on variable sizes of the foundation language model. Results with the prompting approach and comparison to the instruction-tuning technique are in Table 10.

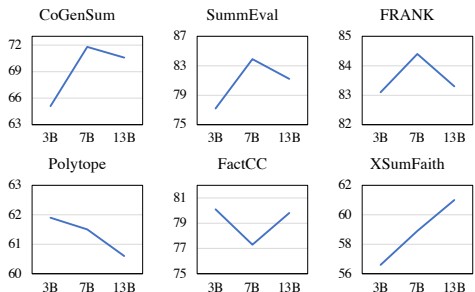

Figure 3: Spearman correlation(%) of FFLM with different model sizes on inconsistency detection datasets.

The observations for inconsistency detection are in line with those for faithfulness ranting. Another finding is that the gap between prompting approaches and FFLM under this setup is much smaller than that under the faithfulness rating setup. This indicates that inconsistency detection is easier than faithfulness rating with less number of answer choices when doing faithfulness evaluation by prompting large models.

## C  A Collection of Prompts

The prompts we collected are in Table 11. We made tiny changes to some of the prompts to enhance their probability of generating an acceptable answer, such as adding a hint like "Marks:" and moving the answer choices to the end of the prompt. We recognize the faithfulness score by analyzing the generations with simple rules. If there isn't an acceptable answer, we regard the summary as unfaithful, i.e., "No" for inconsistency detection and "1" for faithfulness rating.

## D  Performance of Different Prompts

We show the performances of different prompts with Vicuna-7B for inconsistency detection in Table 12 and for faithfulness rating in Table 13. The performance with different prompts for the same task varies a lot. And some prompts used in previous works with ChatGPT failed with smaller models, showing the high-level language understanding

| Metric | FRANKCNN | | | QAGSCNN | | | SummEval | | | FRANKXSUM | | | QAGSXSUM | | |
|---|---|---|---|---|---|---|---|---|---|---|---|---|---|---|---|
| | $\gamma$ | $\rho$ | $\tau$ | $\gamma$ | $\rho$ | $\tau$ | $\gamma$ | $\rho$ | $\tau$ | $\gamma$ | $\rho$ | $\tau$ | $\gamma$ | $\rho$ | $\tau$ |
| CoP | 56.1 | 51.0 | 39.4 | **73.0** | **65.3** | **53.2** | 23.6 | 22.6 | 18.0 | **22.8** | **20.8** | **17.0** | **26.6** | 25.3 | 20.7 |
| HaRiM | **61.0** | 53.9 | 42.1 | 67.4 | 58.2 | 47.1 | **42.7** | **37.6** | **29.8** | 14.8 | 13.9 | 11.4 | 15.8 | 16.0 | 13.1 |
| CoCo$_\text{token}$ | 55.9 | 50.1 | 39.0 | 64.7 | 53.1 | 42.6 | 41.3 | 36.8 | 29.1 | 8.0 | 6.8 | 5.6 | 21.4 | 22.8 | 18.7 |
| CoCo$_\text{span}$ | 56.9 | 50.3 | 39.1 | 66.4 | 56.0 | 45.0 | 39.5 | 35.3 | 27.9 | 12.7 | 11.8 | 9.6 | 25.3 | **26.1** | **21.4** |
| CoCo$_\text{sent}$ | 60.9 | **54.1** | **42.2** | 71.7 | 62.0 | 50.2 | 39.4 | 35.0 | 27.6 | 16.3 | 15.8 | 12.9 | 16.5 | 14.9 | 12.2 |
| CoCo$_\text{doc}$ | 59.9 | 54.0 | 42.1 | 71.6 | 61.9 | 50.2 | 39.1 | 34.8 | 27.5 | 18.5 | 17.2 | 14.1 | 22.1 | 21.5 | 17.6 |

Table 8: Correlations(%) of comparisons among CoP, HaRiM, and CoCo with BART for faithfulness rating. The highest scores are in bold.

| Metric | CoGenSum | SummEval | FRANK | Polytope | FactCC | XSumFaith |
|---|---|---|---|---|---|---|
| FFLM | **71.8** | **83.9** | **84.4** | 61.5 | 77.3 | 58.9 |
| *Ablations on the metric components* | | | | | | |
| $\Delta_Y^{prior}$ | 52.2 | 64.6 | 76.2 | 54.8 | 55.8 | **60.5** |
| $\Delta_X^{prior}$ | 49.5 | 67.9 | 73.4 | 62.0 | 55.4 | 58.6 |
| $\Delta_Y^{cond}$ | 64.4 | 82.9 | 83.1 | 56.8 | 79.0 | 53.5 |
| $\Delta_Y^{prior}, \Delta_X^{prior}$ | 54.7 | 65.6 | 77.4 | 62.0 | 55.0 | 58.9 |
| $\Delta_Y^{prior}, \Delta_Y^{cond}$ | 70.5 | 83.5 | **84.4** | 56.8 | 77.3 | 59.8 |
| $\Delta_X^{prior}, \Delta_Y^{cond}$ | 64.4 | 83.5 | 83.6 | 61.5 | 78.6 | 58.6 |
| *Ablations on the metric design* | | | | | | |
| - w/o $w$ | 69.5 | 83.3 | 83.5 | **66.7** | **77.4** | 57.8 |
| - w/o log | 68.7 | 78.5 | 83.5 | 60.9 | 74.2 | 58.1 |
| - w/o $w$ and log | 65.3 | 80.9 | 83.1 | 64.1 | 75.2 | 56.6 |

Table 9: Ablations of FFLM for inconsistency detection. The highest scores are in bold.

| Metric | CoGenSum | SummEval | FRANK | Polytope | FactCC | XSumFaith |
|---|---|---|---|---|---|---|
| *Prompting Approach* | | | | | | |
| LLaMa-7b | 54.3 | 50.0 | 53.6 | 53.7 | 51.7 | 51.7 |
| Vicuna-7b | 56.9 | 58.1 | 69.2 | 54.6 | 69.0 | 55.5 |
| Alpaca-7b | 57.8 | 50.0 | 57.5 | 52.6 | 58.8 | 51.1 |
| *Our Approach* | | | | | | |
| LLaMa-7b | **71.8** | 83.9 | **84.4** | **61.5** | 77.3 | 58.9 |
| Vicuna-7b | 68.6 | 83.2 | 83.8 | 58.3 | 77.2 | 58.9 |
| Alpaca-7b | 65.2 | **85.0** | 83.9 | 59.1 | **78.5** | **60.7** |

Table 10: Comparisons with prompting and instruction-tuning techniques under the same model size for inconsistency detection. The highest scores are in bold in each column and are underlined among each kind of approach.

and generation ability requirements for prompting
large language models.

| Citation | Prompt |
|---|---|
| *Inconsistency Detection* | |
| Chen et al. (2023) | {Document}
Q: Can the following statement be inferred from the above document? Yes or No?
{Summary}
A: |
| Gao et al. (2023) | Is the sentence supported by the article?
Article: {Document}
Sentence: {Summary}
Answer "Yes" or "No": |
| Luo et al. (2023) | Decide if the following summary is consistent with the corresponding article.
Article: {Document}
Summary: {Summary}
Answer (yes or no): |
| *Faithfulness Rating* | |
| Gao et al. (2023) | Evaluate the quality of summaries written for a news article. Rate each summary on consistency. You should rate on a scale from 1 (worst) to 5 (best).
Article: {Document}
Summary: {Summary}
Marks: |
| Luo et al. (2023) | Score the following summary given the corresponding article with respect to consistency from 1 to 10. Note that consistency measures how much information included in the summary is present in the source article. 10 points indicate the summary contains only statements that are entailed by the source document.
Summary: {Summary}
Source Article: {Document}
Marks: |

Table 11: A collection of prompts."{}" marks placeholders for corresponding contents.

| Dataset | Chen et al. (2023) | Gao et al. (2023) | Luo et al. (2023) |
|---|---|---|---|
| CoGenSum | **56.9** | 49.8 | 49.4 |
| SummEval | **58.1** | 50.2 | 54.5 |
| FRANK | **69.2** | 46.9 | 57.8 |
| Polytope | **54.6** | 51.0 | 52.8 |
| FactCC | **69.0** | 51.7 | 53.8 |
| XSumFaith | 52.2 | 49.6 | **55.5** |

Table 12: Balanced accuracy(%) of Vicuna-7B with different prompts for inconsistency detection. The highest accuracy on each dataset is in bold.

| Dataset | Gao et al. (2023) | Luo et al. (2023) |
|---|---|---|
| FRANKCNN | **17.4** | -5.0 |
| QAGSCNN | **19.8** | 4.6 |
| SummEval | **11.9** | -0.9 |
| FRANKXSUM | **5.5** | -3.6 |
| QAGSXSUM | **8.4** | -3.3 |

Table 13: Spearman correlation(%) of Vicuna-7B with different prompts for faithfulness rating. The highest correlation on each dataset is in bold.