# OpenReview forum: "Zero-shot Faithfulness Evaluation for Text Summarization with Foundation Language Model"
_EMNLP/2023/Conference — EMNLP 2023 Main_

### Official Review · Reviewer_df1Z · 2023-08-03

**Typos Grammar Style And Presentation Improvements:** None as far as I know
**Soundness:** 4

**Excitement:**

3: Ambivalent: It has merits (e.g., it reports state-of-the-art results, the idea is nice), but there are key weaknesses (e.g., it describes incremental work), and it can significantly benefit from another round of revision. However, I won't object to accepting it if my co-reviewers champion it.

**Missing References:**

None as far as I know

**Paper Topic And Main Contributions:**

This paper presents a novel evaluation metric FFLM to solve zero-shot faithfulness evaluation for summarization. FFLM is designed based on the probability output of foundation models. FFLM is a parametric combination of changes with prior probability and changes with conditional probability. Extensive experiments and ablation studies on various benchmarks compared with many recent methods show the effectiveness of FFLM, given a comparatively small backbone model and no fine-tuning required.

**Questions For The Authors:**

Q1: what is the correlation of these metric components in Table 5 experiments?

Q2: Can we adjust the score or weighting parameters by removing the effect of different lengths in the generation prefixes of different component metrics?


**Reasons To Accept:**

1. The guideline of the metric is clearly motivated: independent of other tasks and dataset-specific expertise.

2. This work contains clear explanation on the concepts and comparison with other work. For example, the authors clarify that extrinsic hallucination is unfaithful in this work, which is often ambiguous in other recent work.

3. The comparison and ablation are nicely designed, for example, the comparison between prompting and instruction-tuning is much appreciated.


**Reasons To Reject:**

1.	FFLM is backed on LLaMa 7B. Is it possible to use smaller sized models such as Flan-T5 to conduct the same experiments?

2.	The authors mentioned that the combination weights can be tuned, besides naïve 0.25 0.25 0.5 and all 1/3, mentioned in 4.3. Are there results based on tuning on dev set?


**Reproducibility:**

4: Could mostly reproduce the results, but there may be some variation because of sample variance or minor variations in their interpretation of the protocol or method.

**Reviewer Confidence:**

3: Pretty sure, but there's a chance I missed something. Although I have a good feel for this area in general, I did not carefully check the paper's details, e.g., the math, experimental design, or novelty.

---

> ### Author Rebuttal · Authors · 2023-08-27
>
> Thank you for your precious comments. We will follow your advice to improve the paper.
>
> **Response to Reject 1:**
>
> Yes, it's possible to use smaller-sized models with some prerequisites:
>
> First, the model should be a foundation language model without being fine-tuned on task-specific datasets. The drawbacks of models trained on task-specific datasets are explained in Line 380-391. Flan-T5 was fine-tuned on Muffin corpus containing a large amount of summarization data, thus it's not recommended.
>
> Second, this is highly related to research on investigating the optimal model size and dataset size for training foundation language models as mentioned in Sec. 4.5.
>
>
>
> **Response to Reject 2:**
>
> Yes, the combination weights are tuned on the dev set if it exists. The weights are tuned on the dev set for the SummaC benchmark (Table 2), and set empirically for the faithfulness rating datasets (Table 3).
>
> **Response to Q1:**
>
> In Table 5, we mainly show the ablation study of our metric FFLM, and all of the results are calculated between the faithfulness ratings by human annotators and automatic metrics.
>
> The Pearson/Spearman/Kendall correlations(%) between the pairs of the metric components are as follows:
>
> | | FRANKCNN | QAGSCNN | SummEval | FRANKXSUM | QAGSXSUM |
> |  ----  | ----  |  ----  | ----  |  ----  | ----  |
> | $\Delta_{Y}^{prior}$, $\Delta_{X}^{prior}$ | 44.7 / 46.2 / 32.0 | 32.5 / 35.8 / 23.8 | 29.5 / 34.4 / 23.7 | 28.3 / 31.1 / 20.9 | 28.0 / 24.9 / 17.0 |
> | $\Delta_{Y}^{prior}$, $\Delta_{Y}^{cond}$ | 26.5 / 19.9 / 12.9 | 14.7 / 2.3 / 1.2 | 20.1 / 15.6 / 10.1 | 23.4 / 20.1 / 13.4 | -6.2 / -2.4 / -1.5 |
> | $\Delta_{X}^{prior}$, $\Delta_{Y}^{cond}$ | 47.2 / 49.9 / 34.6 | 35.2 / 41.2 / 28.0 | 36.5 / 40.6 / 27.9 | 38.7 / 38.3 / 25.9 | -0.4 / 3.8 / 2.5 |
>
> We can see that the correlations vary among different datasets, and none of the pairs show a high degree of correlation. This also indicates that these components may capture unfaithfulness from different aspects.
>
> We will add the results of the correlation of these metric components in the Appendix.
>
>
>
> **Response to Q2:**
>
> If I understand you correctly, when you say "the generation prefixes", you mean the generated tokens before each step. For example,
> {$y_0, ..., y_{t-1}$} are the prefix tokens of predicting $y_{t}$ for LM(Y). If that is the case, in this work we simply take the average of them (see Eq. 6) and have achieved favorable results compared with the baselines. It should be noted that we calculate the probability of each token under the teacher-forcing strategy (Line 127-128), which won't lead to the exposure bias issue.
>
> If what you mean is "the length of generation contexts", for example, X is the context of Y in calculating LM(Y|X). We didn't consider the effect of it in the current proposed metric, and will consider it in the future.

---

### Official Review · Reviewer_cG7n · 2023-08-07

**Soundness:** 4

**Excitement:**

3: Ambivalent: It has merits (e.g., it reports state-of-the-art results, the idea is nice), but there are key weaknesses (e.g., it describes incremental work), and it can significantly benefit from another round of revision. However, I won't object to accepting it if my co-reviewers champion it.

**Paper Topic And Main Contributions:**

This paper presents a metric for zero-shot faithfulness evaluation of summaries simply with a foundation language model. The work is very similar to (the same motivation) She et al. (2023) and Son et al. (2022) as cited by the authors with slight changes in the formulation of the actual proposed metric.

**Questions For The Authors:**

Can you provide the main difference to HaRiM conceptually, though the formula is provided in Appendix (and it varies with the formula for the proposed method) from my understanding it seems exactly similar conceptually with an additional constraint.

**Reasons To Accept:**

The paper does an excellent job in terms of comparing with a large number of models, on various datasets and correlating across multiple (currently adopted) metrics. This includes comparing with just the output of LLM without the proposed metric.

The metrics show a strong correlation with human ratings pointing to the usefulness of the proposed methods. Further, the formulation is very intuitive, and all the 3 components of the weighted sum make sense intuitively (along with providing a somewhat naive interpretablity).

**Reasons To Reject:**

The technique proposed is extremely similar to HaRiM, though HaRiM ends up with a more complex formula, the proposed metric only seems to add a back constraint of P(X|Y) additionally to HaRiM's method. Further note, HaRiM_LLaMa in fact outperforms the given metric on human correlation in Table 4.

**Reproducibility:**

4: Could mostly reproduce the results, but there may be some variation because of sample variance or minor variations in their interpretation of the protocol or method.

**Reviewer Confidence:**

4: Quite sure. I tried to check the important points carefully. It's unlikely, though conceivable, that I missed something that should affect my ratings.

---

> ### Author Rebuttal · Authors · 2023-08-27
>
> Thank you very much. We will explain the differences between our work and the baselines more clearly.
>
> First, we apologize that there is a typo in Eq. 9 in the Appendix and it should be:
>
> $HaRiM = \frac{1}{m}\sum_{i=1}^{m}(1-p_{y_i}^{s2s})[1-(p_{y_i}^{s2s} - p_{y_i}^{prior})]$
>
>
> Our metric FFLM is different from HaRiM in three ways:
> 1. We propose to do faithfulness evaluation with foundation language models while they suggested to evaluate with models fine-tuned on summarization dataset. HaRiM_LLaMa is an improved metric with our proposal and performs quite differently from the original HaRiM as discussed in Line 380-391: The original HaRiM tends to prefer summaries generated by abstractive models, while HaRiM_LLaMa doesn't.
>
> 2. HaRiM only considers one of the changes with prior probability, i.e., $\Delta_{p_Y}^{prior}$.  Our metric not only adds a back constrained of P(X|Y) additionally to HaRiM, but also considers the probability changes with conditional probability as explained in Line 168-185.
>
> 3. Specifically for the formula, the core of HaRiM and one of the three components in our FFLM are both $\Delta_{p_Y}^{prior}$. Nevertheless, the intuition of adding weights behind HaRiM and our FFLM are different regardless of the function-form variations.
>     * HaRiM's weight $(1-p_{s2s})$ was originally introduced to adjust the loss scale for training better neural machine translation models by Miao et al.(2021)[1].
>     * In our formula, we adopted the weight $e^{p_{y_i}^{s2s}}$ and the logarithm to pay more attention to low-probability tokens which generally correspond to unfaithful contents according to Goyal et al.(2022)[2].
>
> [1] Mengqi Miao, Fandong Meng, Yijin Liu, Xiao-Hua Zhou, and Jie Zhou. 2021. Prevent the language model from being overconfident in neural machine translation. ACL.
>
> [2] Tanya Goyal, Jiacheng Xu, Junyi Jessy Li, and Greg Durrett. 2022. Training dynamics for text summarization models. findings of ACL 2022.

---

### Official Review · Reviewer_TWe7 · 2023-08-09

**Soundness:** 4

**Excitement:**

4: Strong: This paper deepens the understanding of some phenomenon or lowers the barriers to an existing research direction.

**Missing References:**

Scores of ChatGPT with Chain of Though reasoning are available for the SummaC benchmark, but not reported in this paper.

**Paper Topic And Main Contributions:**

This work is on the faithfulness (factual consistency) evaluation of abstractive summaries. It is an important research direction to improve natural language generation models and reduce their hallucinations. In particular, this work considers the task of zero-shot faithfulness evaluation, in contrast to weakly-supervised methods.

The authors contribute a new evaluation metric FFLM, a zero-shot faithfulness evaluation based on foundation language models (they use Llama). In their experiments, they extensively show that their approach has favorable performance over state-of-the art models, especially ChatGPT-based approaches, even though their model is "small" (7B parameters) in comparison.

**Reasons To Accept:**

- the authors advance a recently introduced (and thus small) line of work: faithfulness evaluation with probability changes
- the proposed FFLM model achieves SotA results on many datasets and is competitive with other approaches on the other evaluated datasets, even though it uses less parameters. Further, their method achieves consistent results across all datasets as opposed to other approaches (even on the harder XSum based datasets), demonstrating good generalization capabilities over dataset.
- the experiments are extensive. Models are evaluated in two important faithfulness tasks "Faithfulness Rating" and "Inconsistency Detection". Further, all components of their FFLM approach (which consists of three measurements of probability changes and two token weighting schemes) are analyzed in ablation studies as well as error analyses.
- the paper is well structured and clearly written, which makes it easy to understand

**Reasons To Reject:**

- the advances over the state-of-the-art are small
(- this work describes incremental work)

**Reproducibility:**

5: Could easily reproduce the results.

**Reviewer Confidence:**

4: Quite sure. I tried to check the important points carefully. It's unlikely, though conceivable, that I missed something that should affect my ratings.

---

> ### Author Rebuttal · Authors · 2023-08-27
>
> We profoundly appreciate the hard work and the precious comments.
>
> **Response to "the advances over the state-of-the-art are small"**
>
> In terms of method innovation:
> * The core of this work is to show the feasibility of doing the zero-shot faithfulness evaluation with a foundation language model (instead of a fine-tuned model or prompting methods).
> * Besides, we classify the probability changes into two categories and show that different probability changes are complementary to each other to some extent, which can achieve further improvements when considering them all together.
>
> In terms of result improvements:
> * With our metric FFLM, a 7B-parameter language model can surprisingly outperform ChatGPT on most of the datasets and perform competitively on the others.
> * FFLM also shows strong generalization ability among different datasets (see Table 1), tasks (inconsistency detection in Table 2 and faithfulness rating in Table 3), and evaluation setups (summary-level in Table 3 and system-level in Table 4), compared with the baselines.
>
> * It should also be noted that the baselines FactCC and SummaC in Table 2 are weakly-supervised methods, while our FFLM is a zero-shot evaluation metric. FFLM can still outperform them on CoGenSum, SummEval, and Frank, while ChatGPT lags behind.
>
> Based on the above two points, we think our work makes considerable improvements over the state-of-the-art.
>
>
> **Response to "Scores for ChatGPT-CoT on the SummaC benchmark":**
>
> We didn't include it because: Chen et al.(2023)[1] found that "chain-of-thought prompting hurts performances dramatically compared to vanilla prompting in most cases" as faithfulness evaluation is less reasoning-intensive. Thus, we only listed the results for ChatGPT without CoT in the paper. This has been mentioned in the footnote on Page 3.
>
> [1] Shiqi Chen, Siyang Gao, and Junxian He. 2023. Evaluating factual consistency of summaries with large language models.

---

### Meta-Review · Area_Chair_zZ1k · 2023-09-17

**Recommendation:** 5

**Metareview:**

The proposed technique to evaluate factuality in a zero-shot manner leveraging probability changes of a moderately sized model (LLama-7B). The reviewers all agree that the proposed method is extensively evaluated and would be useful for the community. The authors should clearly distinguish their approach from prior work (HaRiM) in future versions.

---

### Decision · Program_Chairs · 2023-10-07

**Decision:**

Accept-Main

**Comment:**

The proposed technique to evaluate factuality in a zero-shot manner leveraging probability changes of a moderately sized model (LLama-7B). The reviewers all agree that the proposed method is extensively evaluated and would be useful for the community. The authors should clearly distinguish their approach from prior work (HaRiM) in future versions.